# Assembling of Carbon Fibre/PEEK Composites: Comparison of Ultrasonic, Induction, and Transmission Laser Welding

**DOI:** 10.3390/ma15186365

**Published:** 2022-09-13

**Authors:** Adrian Korycki, Christian Garnier, Margot Bonmatin, Elisabeth Laurent, France Chabert

**Affiliations:** 1Laboratoire Génie de Production, ENIT-INPT, University of Toulouse, 47 Avenue d’Azereix, 65016 Tarbes, France; 2Institut Clement Ader (ICA), University of Toulouse, CNRS, IMT Mines Albi, INSA, ISAE-SUPAERO, UPS, Campus Jarlard, 81013 Albi, France; 3CNES, Sous-Direction Assurance Qualité, Service Technologies, Matériaux et Procédés, 18 Avenue Edouard Belin, 31401 Toulouse, France

**Keywords:** thermoplastics, composites, laser welding, induction welding, ultrasonic welding

## Abstract

In the present work, an ultrasonic, an induction, and a through transmission laser welding were compared to join carbon fibre reinforced polyetheretherketone (CF/PEEK) composites. The advantages and drawbacks of each process are discussed, as well as the material properties required to fit each process. CF/PEEK plates were consolidated at 395 °C with an unidirectional sequence and cross-stacking ply orientation. In some configurations, a polyetherimide (PEI) layer or substrate was used. The thermal, mechanical, and optical properties of the materials were measured to highlight the specific properties required for each process. The drying conditions were defined as 150 °C during at least 8 h for PEI and 24 h for CF/PEEK to avoid defects due to water. The optical transmission factor of PEI is above 40% which makes it suitable for through transmission laser welding. The thermal conductivity of CF/PEEK is at most 55 W·(m·K)^−1^, which allows it to weld by induction without a metallic susceptor. Ultrasonic welding is the most versatile process as it does not necessitate any specific properties. Then, the mechanical resistance of the welds was measured by single lap shear. For CF/PEEK on CF/PEEK, the maximum lap shear strength (LSS) of 28.6 MPa was reached for a joint obtained by ultrasonic welding, while an induction one brought 17.6 MPa. The maximum LSS of 15.2 MPa was obtained for PEI on CF/PEEK assemblies by laser welding. Finally, interfacial resistances were correlated to the fracture modes through observations of the fractured surfaces. CF/PEEK on CF/PEEK joints resulted in mixed cohesive/adhesive failure at the interface and within the inner layers of both substrates. This study presents a guideline to select the suitable welding process when assembling composites for the aerospace industry.

## 1. Introduction

Polymer welding is a process of assembling surfaces of thermoplastic-based materials, generally with heat and under pressure [1,2,3]. Only thermoplastics of the same nature or miscible with each other can be assembled by welding. Welding is a relevant assembling process to contribute to the reduction of the environmental impact of the lightening of structures. Besides, as polymer welding is fast, safe, and cheap, spreading such a process in the industry will contribute to the competitiveness of companies. Welding thermoplastics is quite well mastered, whereas more research is required to target welding of short and long-fibre thermoplastic composites. Moreover, some issues remain to make the shift toward welding complex shapes and large structures of thermoplastic polymers and composites.

A choice of a process is affected by materials to be joined, a joint configuration, a required mechanical strength, a level of seal, process costs and speed, and production volumes. However, the creation of a good quality weld does not only depend on the welding process but also on the weldability of the materials to be joined [4]. Therefore, the evaluation of their weldability is of high importance in the whole welding operation for a successful assembly of polymer materials.

Compared to thermosets, thermoplastics can be recycled and do not require refrigerated storage, offering almost an infinite shelf life. Most of the thermoplastic composites used in marine, aerospace, and automotive lightweight structures are made of carbon fibre reinforced polymers (CFRP) composites, which give way to glass fibre reinforced polymers (GFRP) for some applications. Carbon fibres give better properties than glass fibres because of their higher stiffness and strength. Sometimes, the carbon fibres used are high modulus (HM) for the highest performance required in some applications. The glass or carbon fibres associated with a matrix of whether polyphenylene sulfide (PPS), polyetherimide (PEI) or polyaryletherketone (PAEK), mostly polyetheretherketone (PEEK) and polyetherketoneketone (PEKK) [3,5]. These high-performance polymers display high damage tolerance to finished parts, as well as chemical resistance, and, thus, are stable in severe hot or wet conditions. Thermoplastic composites can be re-melted with promising benefits in repairing and end-of-life re-using. Thermoplastics are a response to the growing demand for new materials that can be used in aircraft structures with a reduced total weight while maintaining high mechanical properties. In addition, despite the high price of thermoplastics such as polyetherketone (PEK), PEEK, and PPS, the total cost of part manufacturing is lowered as a result of a significant reduction in production times.

Moreover, thermoplastic composites offer the ability to save weight and improve the sustainability of airplanes and spacecrafts by joining components via fusion bonding. Welding is an attractive alternative to conventional methods, such as mechanical fastening and adhesive bonding to join composite parts. However, since composites gain their outstanding properties from the fibres, the weld is inevitably the weakest point in the system as the weld is a zone free of fibres.

The welding processes are mainly focused on a couple of advanced materials, such as carbon fibre reinforced polyetheretherketone. Also, glass and carbon fibre reinforced polyetherimide (GF/PEI and CF/PEI) have been investigated. CF/PEEK is in most cases the APC-2, a semi-product manufactured by Imperial Chemical Industries (UK), while PEI-based materials are known as Cetex, supplied by Ten Cate Advanced Composites (The Netherlands). Some studies were also attracted to polyphenylene sulphide or polypropylene (CF/PPS or GF/PP and CF/PP) [6]. 

Many welding techniques have been proposed, developed, and evaluated for thermoplastic applications. Based on the mechanism of heat generation at the welding interface, welding methods for thermoplastics can be classified as external (thermal heat source) and internal (mechanical movement or electromagnetism) heating methods. They differ in the way heat is generated at the interface, such as frictional heating (e.g., ultrasonic welding [7,8,9,10]), electromagnetic heating (e.g., induction welding [11]), and thermal techniques (e.g., laser welding [12,13,14,15,16,17]). The selection of the welding method depends mainly on the thermal and electrical properties of the material and the shapes of the elements to be joined [18]. Ultrasonic, induction, and laser welding are the most used processes to assemble such thermoplastic composites.

Whatever the process, the physical mechanisms during welding of thermoplastics are accomplished in three sequential stages: a surface preparation, an application of heat, and a weld creation on cooling. Upon pressure, the surfaces come into close contact which initiates a macromolecular diffusion and an entanglement across the joint. Only thermoplastics and thermoplastic-matrix composites can be welded [19]. Compared to thermosets, their chains are free to move up to long distances when intermolecular weak bonds are broken. All the welding processes are governed by the same parameters: heat, pressure, and time [20]. To achieve high-quality welds, a careful optimization of the welding parameters is required for each application. 

In ultrasonic welding, a sonotrode coupled to a transducer that produces high frequency (20–40 kHz) vibration causes frictional heat and melting at the interface of two parts. A polymeric film is required at the interface as an energy director. Among the results published until now, studies on high-performance thermoplastic composites include CF/PEI [21,22], CF/PPS [23,24,25], and CF/PEEK [10]. The strength of joints in ultrasonic welding has been modeled and reported by Benatar and Gutowski [26] for PEEK/AS4 graphite APC-2 composites. The strength of the welded parts obtained was estimated as 74 MPa. The effect of welding time was investigated by Tao et al. [10] for CF/PEEK with a flat PEEK film as an energy director. It was reported that with a gradual increase in welding time, the weld strength also increased up to an optimum time of 0.9 s, the maximum LSS obtained was 28 MPa. Longer welding time leads to overheating and local degradation [10]. Recently, CF/PEEK assembled by an ultrasonic welding reached a maximum LSS at 49 MPa [27]. The same authors provided thermal profiles during welding from microthermocouples located close to a welded interface. The evolution of the interfacial temperature during welding was correlated to welding parameters to prevent overheating. Moreover, a continuous ultrasonic welding has been currently under development [7,28,29]. 

In induction welding, an electromagnetic coil produces alternating electromagnetic fields which induce Eddy currents in conductive laminate. Eddy currents produce heat through carbon fibres or a metallic susceptor which melts the thermoplastic. The perpendicular 0° and 90° fibre orientation in the woven fabric are ideal, enabling Eddy currents to be generated in each ply of the laminate. With unidirectional laminated stacks, however, it is common to have 45° plies interspersed so that the angle difference is smaller. Williams et al. [30] by using a woven fabric susceptor made from co-spun fibre yarn comprising blended staple carbon and PEEK fibres welded APC-2 laminates. An LSS of 46 MPa was obtained. However, induction welding without a susceptor required the electrical and thermal conductivity of the materials to fit the process. The main consideration for design was how to control and concentrate the magnetic field onto the workpiece. Using no susceptor, Cogswell et al. [31] were able to obtain an LSS of 31 MPa for APC-2 laminates joined. Border and Salas [32] investigated induction welding on PEEK without metal susceptors. An LSS of 48.2 MPa was obtained. Reis et al. [3] review presented that the effectiveness of induction welding in several thermoplastic composites was evaluated with lap shear strength values ranging between 14 MPa and 43 MPa. 

During the transmission-through laser welding process, laser radiation passes through an upper laser transparent part, for instance, unreinforced amorphous PEEK and it generates heat in a lower laser absorbent part which contains optionally carbon fibre or conductive additive. Amanat et al. [33] welded PEEK films using a pulsed fibre laser (1060 nm) and a maximum power of 20 W. According to their results, the two lowest scan speeds, 4 mm·s^−1^ and 8 mm·s^−1^, showed the most significant bond strengths of around 22 MPa to 25 MPa for semi-crystalline and 12 MPa to 19 MPa for amorphous PEEK. Since carbon fibres are laser absorbent, CF/PEEK is suitable as a lower part only. The reliable application of laser welding technology for joining primary aeronautic parts made of thermoplastic composites still requires significant development and investigation [15]. Torrisi et al. [34] used 3-ns Nd: Yag laser (532 nm) system with various fillers for ultra-high molecular weight polyethylene composites. They found that the best polymer coupling was with carbon nanotubes which showed a good adhesion value with 10 MPa shear rupture at an irradiation time of 2 min.

Ultrasonic, induction, and laser welding processes are already widespread for assembling pure thermoplastics. However, welding fibre-reinforced composites bring new issues to be tackled compared to pure thermoplastics. Indeed, the fibres modify the properties of the materials, mainly optical, thermal conductivity, and rheological properties. As a consequence, all the process parameters need an overhaul to reach repetitive and reliable welds.

The purpose of this study is to compare three welding processes for assembling high-performance thermoplastic composites for spacecraft applications. Whereas each existing study focuses on one process/material couple, the originality of the hereby article is to compare three welding techniques to join the same materials. Ultrasonic welding (UW), induction welding (IW), and transmission laser welding (TLW) are applied to assemble CF/PEEK thermoplastic composites and PEI on CF/PEEK. The materials are characterized to measure their water intake, thermal transitions, thermal stability, thermal conductivity, thermomechanical properties, and optical properties. Then, the mechanical resistance of welds is measured by single lap shear tests. Finally, the main strengths and flaws of each process are highlighted as well as the material requirements for successful welding of CF/PEEK composites.

## 2. Materials and Methods

### 2.1. Materials and Samples Preparation

The materials used in this study are:Thermoplastic composite prepregs carbon fibre HM63/PEEK developed by Suprem (Yverdon-les-Bains, Switzerland) and supplied by Centre National d’Etudes Spatiales (CNES, Toulouse, France) as 140 µm thick tapes. The latter, named CF/PEEK in the following, was made of PEEK 150G from Victrex as a matrix and unsized carbon fibre HM63 from Hexcel. HexTow HM63 carbon fibre is a continuous, high strength of 4.5 GPa and high modulus of 452 GPa, with a density of 1.83 g·cm^−3^, polyacrylonitrile-based carbon fibre.PEEK 450 G from Victrex (Lancashire, UK) in granules for DMTA and TGA only.PEI Ultem 1000 from Sabic (Rijad, Saudi Arabia) in granules.PEI Ultem 1000 as 250 µm thick film was purchased from GoodFellow (Lille, France).

CF/PEEK, pure PEEK, and PEI were processed by compression moulding. A hydraulic press LAB 800P PEI from Pinette Emidecau Industries (Chalon-sur-Saône, France) was used for preparing 1, 2, and 4 mm thick plates. The materials were dried in a vacuum oven at 150 °C for at least 3 h before processing. The plates of CF/PEEK 1 mm with a unidirectional sequence of the tapes [0°_7_] (Figure 1a) and plates with cross stacking [45°,−45°,0°_3_,−45°,45°] (Figure 1b) were consolidated at 395 °C and 2 MPa for 30 min, the heating, and cooling rates were 4 °C·min^−1^. The unidirectional sequence was expected to dissipate heat faster and more efficiently. This configuration is the worse for fusion bonding because most of the generated heat is dissipated on each side of the specimen. Higher temperatures must be attained to melt the interface to be welded. The cross-stacking sequence is close to industrial cases and it was adapted to fit the induction welding process by adding a 45° layer on the upper surface of the specimens. PEEK pellets were kept in a cavity of 2 mm thickness between two steel foils, and the temperature was increased at a speed of 10 °C·min^−1^ up to 360 °C. A pressure of 10 MPa was applied at 360 °C for 5 min. Then, still under pressure, the moulded samples were cooled down to 200 °C at the speed of 4 °C·min^−1^ before demoulding. PEI plates with a nominal thickness of 1, 2, and 4 mm were produced from granules at 290 °C and 10 MPa for 5 min, and heating and cooling rates were 10 °C·min^−1^. The plates with dimensions of 100 × 50 mm^2^ (1 mm thick), 150 × 75 mm^2^ (2 mm thick), and 110 × 75 mm^2^ (4 mm thick) were cut into samples of 100 mm long and 25 mm wide.

### 2.2. Characterization of Materials

During welding, between the two elements to be joined, diffusion and interface crossing of macromolecules take place. The physical phenomena of macromolecular diffusion, healing, and crystallization at the interface are well described in Martineau’s work [35] for PEEK. 

After cooling, a permanent cohesive joint is formed. The appropriate mobility of the diffusing chains is reached when the temperature is above the melting temperature for semi-crystalline polymers or well above glass transition for amorphous polymers. Also, the materials to be joined should have a similar softening or melting point, must be miscible, or must have a similar chemical structure [36]. Moreover, thermoplastics are sensitive to thermo-oxidative degradation. During welding, overheating can occur and it weakens the interfacial strength [37].

To understand the interactions of the welding parameters with the specimens, careful consideration of the material properties is beneficial. Main properties were identified as playing a role in the welding result and divided into two groups: (1)roughness and surface chemistry,(2)rheological, optical, and thermal properties.

For the sake of brevity, only the second group is presented in this article. Moreover, the presence of water within the specimens has been proved to be detrimental to the joint integrity and mechanical resistance of welds [38]. For this reason, careful drying is necessary before assembling the specimens. Also, the water content could influence further material characteristics.

#### 2.2.1. Drying of Specimens

PEEK and PEI are sensitive to water because they contain chemical groups such as ketones, ethers, and hydroxyls which create weak bonds with H_2_O. So, a thorough study of the kinetics of sorption and drying was carried out to determine an appropriate drying procedure. Our approach consists of three steps:(1)drying at 150 °C for at least 3 h,(2)immersion of the specimens in a beaker containing deionized water at 25 °C. A thermocouple IKA ETS-D5 from KA-WERK (Brachbach, Germany) was placed inside the beaker and the latter was maintained on a hotplate stirrers IKA C-MAG HS7 from KA-WERK (Brachbach, Germany) to keep a constant temperature throughout the experiment,(3)the specimens were placed in an oven Memmert UNB 200 (Schwabach, Germany) at 150 °C. Mass readings with a Mettler Toledo scale (Viroflay, France) during sorption and drying were taken over a short time interval at first every hour, then twice a day.

Besides the evolution of weight with time, the presence of water inside the materials was checked by Fourier Transform Infrared Spectrometry. Spectra were recorded using an FTIR spectrometer Spectrum One by Perkin Elmer (Waltham, MA, USA) in attenuated total reflectance (ATR) mode in the 4000–650 cm^−1^ range. The resolution was 4 cm^−1^, and 16 scans were accumulated for an improved signal-to-noise ratio.

#### 2.2.2. Dynamic Mechanical Analysis

Thermomechanical properties were measured by dynamic mechanical thermal analysis (DMTA) using ARES LN2 rheometer from TA Instruments (New Castle, DE, USA) in torsion mode. Parallelepipedic specimens of 45 × 10 × 2 mm^3^ were cut out from moulded plates. Temperature ramps from 25 °C to 325 °C at a heating rate of 3 °C·min^−1^ were applied at a frequency of 1 Hz and strain of 0.1% within the linear viscoelastic (LVE) domain of both PEEK and PEI. Previously, strain sweeps were carried out to define the LVE domain.

#### 2.2.3. Optical Properties

Optical properties are very important for transmission laser welding. Transmission factors of PEEK and PEI were measured with a VERTEX 70 spectrophotometer from Bruker (Billerica, MA, USA) in the spectral range between 400 nm and 20.000 nm. The transmission mode with an integrating sphere and an incident angle of 0° between the beam and the normal to the sample was applied. The tests were carried out at room temperature with the integrating sphere. The evolution of the transmittance as a function of temperature was measured between 20 °C and 140 °C with an increment of 20 °C.

#### 2.2.4. Thermal Properties

Only thermoplastics whose processing temperatures are in the same range can be welded together. Both parts must be melted to allow the diffusion of macromolecules through the interface. Thermal transitions of PEEK and PEI were measured by differential scanning calorimetry with a DSC 1 Star System from Mettler Toledo (Viroflay, France). The measurements were performed under a nitrogen flow of 50 mL·min^−1^, and a mass of approximately 10 mg was placed in sealed aluminium pans Mettler Toledo (Viroflay, France). The DSC scan comprised two steps: (1)an ascending temperature sweep of 10 °C·min^−1^, from 25 °C to 400 °C,(2)a descending temperature sweep of 10 °C·min^−1^, from 400 °C to 25 °C.

Glass transition temperature (T_g_), crystallization temperature (T_x_), and melting temperature (T_m_) were obtained. 

Then, the behavior of materials under high-temperature conditions was evaluated through a TGA 2 Mettler Toledo device (Viroflay, France). The degradation temperature (T_d_) was obtained using platinum crucibles Mettler Toledo (Viroflay, France) under an oxygen atmosphere at 50 mL·min^−1^, sample mass of 10–15 mg, and the heating rate at 5 °C·min^−1^ in the range from 25 °C to 800 °C. 

Thermal conductivity and specific heat were measured by using Hot Disk TPS 2500 S analyzer (Göteborg, Sweden). The hot disk probe was placed between two 5 mm thick samples of the material to be characterized. Surfaces of the samples in contact with the probe are flat. The principle is to pass an electric current through the probe to generate a temperature increase of one to several degrees and to record the increase in resistance (temperature) over time. The temperature of the measurement was 23 °C.

#### 2.2.5. Degree of Crystallinity

The mass fraction of the crystalline phase was calculated from the melting enthalpy measured by DSC and the theoretical melting enthalpy of 100% crystalline phase with the following formula:(1)X=∆Hmω
PEEK·
1∆Hth·100 [%]
where *X* is the degree of crystallinity [%], Δ*H_m_* is the melting enthalpy [J·g^−1^], *ω_PEEK_* is the mass fraction of PEEK of the composite and Δ*H_th_* is the theoretical melting enthalpy of 100% crystalline phase of PEEK which is 130 J·g^−1^ [39].

### 2.3. Welding Processes

#### 2.3.1. Ultrasonic Welding

One-mm thick unidirectional plates of CF/PEEK and 1mm thick PEI plates were used. Single-lap welded joints with the two different types of welding stacks sketched in Figure 2 were considered in this study. The first configuration was a direct weld between PEI and CF/PEEK. The second configuration consisted of two CF/PEEK plates with a PEI film as an energy director. It was a 250 µm thick flat energy director, which was placed at the welding interface before the welding process. It concentrated heat generation at the welding interface through combined surface friction, i.e., friction between the energy director and composite plates moving relative to each other, and viscoelastic friction, i.e., friction among adjacent polymer molecules in the energy director when subjected to cyclic deformation. This process provided welded joints over an entire overlapped surface. The specimens were welded in a single lap configuration (according to ASTM D1002) with a 10 mm long overlap.

Electrical Motion 20 ultrasonic welder from Rinco Ultrasonics (Romanshorn, Switzerland) was used for the experiments. An ultrasonic welding equipment was used to transmit mechanical vibration at a high frequency to the joint interface along with a static compressive force. The working frequency is 20 kHz and the maximum load of 3000 N. Amplifications provided by the booster and the titanium sonotrode were, respectively, 2 and 4. A clamping tool was designed to provide vertical movement without parasite bending to ensure pure friction during the welding test. Three welding parameters were considered in this study: the welding load of 500 N, the vibration amplitude of 32 µm, and the welding time of 1000 ms. The welding time is associated with the time when the ultrasound is operating. The welding load continued to be applied during the selected 2500 ms cooling phase.

#### 2.3.2. Induction Welding

1 mm thick plates with cross stacking of CF/PEEK and 1 mm thick PEI plates were used. Single-lap welded joints with the two different types of welding stacks sketched in Figure 3 were considered in this study.

In the case of PEI on CF/PEEK welds, the inductor was placed 3 mm above the area to be welded. A power of 55–70% and movement of the tool along the weld line at a speed of 2 mm·s^−1^ were used. During the welding of two CF/PEEK plates, the inductor was placed 2 mm above the surface of the welded area. The assembly cycle consisted of gradually increasing the power of the inductor to reach T_m_ + 20 °C at the start of the weld zone. For this purpose, a power of 80% and movement of the tool along the weld line at a speed of 3 mm·s^−1^ were used. 

The welding tool consists of a CEIA 17 kW head (Paris, France) fitted with a 30 mm diameter pancake type inductor located at 2 mm or 3 mm. The inductor was preceded and followed by compaction rollers mounted on pneumatic cylinders. Two orientable Vortex fans (Terrebonne, QC, Canada) allow surface cooling of the weld area or its perimeter. The welding tool is mounted on an ABB IRB6620 6-axis robot (Zurich, Switzerland). The specimens were positioned on top of each other with a 20 mm long overlap and held together with Kapton adhesive. The orientation of the folds at the weld was the same. N-type thermocouples were placed at the start and end of the weld to monitor the temperature of the interface. An induction welding was conducted without a susceptor at the interface, with carbon fibres replacing a metallic insert. An induction welding without a susceptor requires 45° lay-ups on the surface and depends on the electrical and thermal conductivity of the carbon fibre specimens. Induction welding was performed at IRT Saint Exupéry, Toulouse, France.

#### 2.3.3. Transmission Laser Welding

Unidirectional plates of CF/PEEK 1 mm thick, PEI 1, 2, and 4 mm thick plates were used. Single-lap welded joints are sketched in Figure 4. The specimens were positioned on top of each other with a 20 mm long overlap and were held in position employing a pressure of 3.8 MPa applied through a glass plate to the overlap zone. Through transmission laser welding depends on the optical properties of the upper material. 

PEI on CF/PEEK assemblies was made with a welding machine ES Weld 2000 by ES LASER (Bordeaux, France) equipped with an infrared laser beam at a wavelength of 970 nm with a maximum power of 225 W. The ovoid shape of the beam, about 5 mm wide, was obtained by a “top hat” lens allowing a uniform distribution of the laser radiation on the area to be welded. Two scans of the laser beam were arranged at an interval of 5 mm from each other. The laser speed was set up at 17 mm·s^−1^ and was carried out over the entire width of each assembly.

During these tests, the laser power was set between 20% (~34 W) and 35% (~69 W), and the laser scanning speed was between 17 mm·s^−1^ and 33 mm·s^−1^. Visually, the welds were homogeneous for a scanning speed of 17 mm·s^−1^ and a power of 25% (~45 W), 30% (~52 W), and 35% (~69 W) for respectively the 1, 2, and 4 mm thick PEI samples. Then, the power used for each PEI thickness was adjusted more precisely: 45 W for 1 mm thick PEI, 50 W for 2 mm, and 66 W for 4 mm thick.

### 2.4. Characterization of Weld Assemblies

#### 2.4.1. Mechanical Properties 

The weld resistance was measured by the single lap shear test, where the maximum force was obtained after applying a tensile load to the specimens. The welds and test were performed following the ASTM D1002 standard, with 100 × 25 mm^2^ samples corresponding to long and wide, respectively, and a minimal overlap area of 10 × 25 mm^2^, on an Instron universal testing machine at 2800 N·min^−1^ cross-head speed. The lap shear strength was then calculated by Equation (2).
(2)$$LSS=\frac{F_{max}}{A}\;\left[MPa\right]$$

In the lap shear test, the strength of a bonded joint (shear stress) depended on the load to the point of fracture (*F_max_*) and overlapping area (*A*).

#### 2.4.2. Optical Observation 

The internal quality of the joints was inspected through a high-resolution microscope Keyence VHX-6000S (Bois-Colombes, France) in a reflection and transmission mode. Three-dimensional imaging was used for mapping each fractured interface after the mechanical test. Additional images were obtained from a camera Huawei P40 (Shenzhen, China) for a larger visual field. Image analysis of the welded surfaces was carried out using ImageJ v1.51k software (Wayn Rasband National Institutes of Health, Bethesda, MA, USA).

## 3. Results and Discussion

### 3.1. Material Properties

Both PEI and PEEK are sensitive to water intake, which affects the quality of the weld. If water is present, it evaporates upon heating, inducing material swelling and defects. Potente et al. [38] welded undried and pre-dried PEEK, they noticed bubbles in the undried PEEK welds only. Besides, heat damages were reported [33] on amorphous PEEK at a power of 20 W with a focal plane speed of 4 mm·s^⁠^^−1^. Although the authors agree about the role of water on defects, the effect of process parameters, such as power and sample speed, are modified with water content. The combined effect of water and fast heating ramp looks to worsen the apparition of defects. Welding processes are very energy-intensive and heating ramps are very high, possibly higher than 1000 °C·s^−1^. Moreover, in transmission laser welding, heating is precisely located in a line or small points. Therefore, the presence of water in the materials causes unsymmetrical swelling of the substrates or locally, the creation of trapped bubbles due to the vaporization of water during welding.

Figure 5 presents the mass evolution for the sorption and desorption (drying) of PEI plates. Figure 5a shows the stabilization of the water content after 312 h of immersion. The corresponding saturation uptake was 1.30%wt. Then, the drying step in Figure 5b verifies the time required to remove water from PEI. During the first hours of drying, the weight decreased to −1.25%. The weight reduction was slower during the next hours, with an equilibrium after 8 h at a total mass loss of 1.40%. This weight loss was higher than the initial water uptake of 1.30%wt. This is due to the uncompleted drying of the initial state. As explained in Section 2.2.1, the first step was drying at 150 °C for 3 h: this duration was not enough to reach a fully dried state. Finally, our water equilibrium concentration is similar to Merdas’ result of 1.39%wt. at 20 °C to 1.50%wt. at 100 °C [40].

Similarly, the evolution of the weight loss for CF/PEEK plates is presented in Figure 6. Figure 6a shows the diffusion of water into the CF/PEEK plates to achieve water saturation after 600 h (25 days) for a maximum mass gain of 1.17%wt. Figure 6b presents the drying curve of CF/PEEK: a rapid mass loss took place during the first hour and then the mass loss was slower. The equilibrium was reached after 24 h for a total mass loss of 1.24%. Grayson and Wolf [41] measured a water intake of 0.44%wt. at 35 °C to 0.55%wt. at 95 °C.

The chemical analysis of dried and wet samples was performed using Fourier-transform infrared spectroscopy which is based on the absorption of infrared radiation by bonds. The position and shape of absorption peaks in a spectrum indicate the nature of chemical groups characterizing the material. The spectra are provided as Appendix A. For PEI, no structural change or degradation was observed after 24 h of sorption or drying. A slight change appeared after 96 h of drying when the -CH_2_ groups were more visible in the 2700 cm^−1^ and 2750 cm^−1^ bands. Correa et al. [42] followed the evolution of stretching bands of -OH groups with sorption time to determine the dynamics of the transport of water molecules within PEI. He revealed different types of hydrogen bonding. He assigned the sharp peaks at 3655–3562 cm^−1^ to isolated water molecules interacting via H-bonding with the PEI backbone. A second doublet at 3611–3486 cm^−1^ was associated with water molecules self-interacting with the first shell species through a single H-bonding (self-associated or second-shell water molecules).

In the case of CF/PEEK, after 24 h of sorption, a strong stretch of -OH groups appeared at 3390 cm^−1^. As sorption progresses, the spectrum was substantially similar with the intensity of the absorption peak increasing with water uptake. After 24 h of drying, the -OH groups disappeared, which means that water was reversibly removed from the material. Thus, for both materials, the heating time or water exposure was not sufficient to cause high levels of structural modification or chain breakage.

Based on these results, the drying conditions were defined as at least 8 h for PEI and 24 h for CF/PEEK. 

The DSC and TGA thermograms of PEI and CF/PEEK are presented in Figure 7 and Figure 8, respectively. The thermal transitions such as glass transition (T_g_), melting (T_m_) and crystallization temperature (T_x_) were obtained from DSC. TGA thermograms give weight loss when the temperature increases. As a convention, the degradation temperature (T_d_) was considered when the polymer lost 5% of its mass.

The T_g_ of PEI was observed between 211 °C and 216 °C. This reflects the change from the glassy state to the rubbery state in which the polymer chains gain mobility. The degradation took place in two main steps up to a complete weight loss. The degradation temperature was 531 °C. This temperature should not be exceeded to avoid damaging the chemical structure. Augh et al. [43] related no changes in molecular weight, glass transition temperature, or mechanical properties of PEI during an induction process. Because the time scale in the induction heating process was short, almost no degradation was observed. Amancio Filho et al. [44] indicated that the primary thermal degradation mechanism of PEI is chain scission, with a very small decrease in average molecular weight. This minor decrease in molecular weight could be considered irrelevant for the mechanical performance of the joints because it is either above or within the molecular weight range where the strength of the PEI is independent of the variations in this property. So, we assume that the PEI macromolecular structure was not significantly modified by the welding cycles.

For CF/PEEK, the T_g_ of PEEK was estimated at 163 °C. The glass transition of pure PEEK is usually near 150 °C [45]. A slight modification of the macromolecules could stem from the interaction of PEEK macromolecules with carbon fibre surfaces [46]. The maximum melting peak was 343 °C. After melting, CF/PEEK crystallized on cooling with a maximum crystallization peak at 298 °C. The melting enthalpy was measured at 15 J·g^−1^ on heating. Considering that the volume fraction of carbon fibre is 61% of the composite, it was possible to calculate the degree of crystallinity with Equation (1). The degree of it for PEEK was 30% which is lower than the highest values reported in the literature for pure PEEK, up to 40% [45]. However, carbon fibres, apart from crystallization initiation and nucleation that decrease the process duration, do not modify it in general [47]. 

In Figure 8b, the degradation temperature for CF/PEEK is 571 °C, while it is faster for pure PEEK and starts at 556 °C. The degradation of CF/PEEK composites occurred in three steps. When reaching 800 °C, the weight loss was only 55% due to the presence of carbon fibres whose thermal stability is much higher than those of polymers. The thermal degradation of PEEK and CF/PEEK composites was well described in Gaitanelis’ work [48]. 

For welding, the interface temperature must reach the temperature at which molecular mobility allows the interdiffusion of the polymeric chains. For amorphous thermoplastics, the welding temperature is after the glass transition and most often it is necessary to reach T_g_ + 140 °C [49] for the polymer to flow sufficiently. For semi-crystalline thermoplastics, welding takes place at the melting temperature or a few degrees above T_m_ + 70 °C [49], while preventing material degradation. 

Based on these results, the theoretical welding range could be defined between 350 °C (T_g_ of PEI + 140 °C) and the degradation temperature of PEI: 350 °C < T < 530 °C for PEI on CF/PEEK assemblies. The lowest temperature limit, 350 °C corresponds also to the T_m_ of CF/PEEK.

The thermal conductivity of PEI was 0.20 W·(m·K)^−1^, and the specific heat was 1068 J·(kg·K)^−1^. Comparatively, the through-plane thermal conductivity obtained for CF/PEEK was 15.95 W·(m·K)^−1^ and up to 55.00 W·(m·K)^−1^ in plane one. Thus, PEI acts as a thermal insulator in welding, keeping heat inside the material. Oppositely, in CF/PEEK, the fibres contribute to fast heat dissipation inside the whole elements to be welded. The thermal conductivity of CF/PEEK is higher than those of PEI, so the heat is dissipated faster in composites [50]. Regarding induction welding, the fibres play the role of a conductive element, allowing for the generation of Eddy currents inside the composite instead of a metallic susceptor. Rudolf et al. [51] reported that continuous carbon fibres may act similar to closed loops within the composite structure.

The dynamic mechanical analysis gives information on the macromolecular mobility and the temperature processing window. The welding configurations were either CF/PEEK on CF/PEEK, PEI on CF/PEEK and CF/PEEK on CF/PEEK with a 250 µm thick interfacial PEI film. 

Polymers are immiscible in most cases, but PEEK and PEI are fully miscible according to Hsaio [36]. This makes PEI a relevant option to assemble PEEK composites. 

In Figure 9a, the storage modulus (E’) of PEEK shows three distinct regions: a high modulus glassy region where the segmental mobility is restricted, a transition zone where a substantial decrease of E’ with the increase in temperature, and a rubbery region (the flow region) where a drastic decay is seen.

The E’ curve of PEI shows the typical behavior of an amorphous polymer, which migrates from an energy-elastic to an entropy-elastic state after reaching the glass transition temperature. The amorphous plateau was not visible: above 250 °C, the PEI was so soft that measuring is not possible in this configuration. Instead, a plate configuration would be suitable. For both materials, the loss modulus (E’’) curves demonstrate broad and asymmetrical peaks related to the energy dissipated through inter macromolecular frictional motion.

Figure 9b indicates the mechanical response of the glass transition (T_α_) when tan(δ) reaches a maximum. The peak of PEEK was broader and less symmetrical than those of PEI. The peak shape is correlated with the polydispersity and the various relaxation times characterizing macromolecular mobility.

The storage modulus of PEI dropped right after its glass transition at 217 °C, giving the macromolecules a high level of mobility while PEEK chains kept their rigidity. So, PEI would facilitate the motion and the diffusion of PEEK chains across the interface. Thus, less energy would be required to soften the PEEK matrix. Based on these results, PEI could be assembled with PEEK from 250 °C when the PEI macromolecules gained enough mobility to diffuse within PEEK.

For PEEK/PEEK welding, previous studies demonstrated that it is necessary to attain its melting temperature close to 360 °C for effective interfacial bonding [35,52,53].

The transmission coefficient of PEI and CF/PEEK was measured to check their weldability in laser through-transmission configuration in the wavelength range of 400 nm to 1000 nm. For a fixed temperature, the transmittance is influenced by the chemical structure, surface roughness, angle of incidence of the light, and sample thickness. The measurements were carried out at 20 °C for 1, 2, and 4 mm PEI plates as well as for CF/PEEK plates in Figure 10a. The evolution of the transmittance with temperature for PEI was followed in the range of 20–140 °C in Figure 10b.

The transmission of PEI was negligible close to 400 nm, then it increased to 57% for 1 mm thick at a maximum of 1000 nm. At the wavelength of interest at 970 nm, the transmittance was 52%, 50%, and 43% at 20 °C for 1, 2, and 4 mm thick samples, respectively. As expected from the Beer-Lambert law presented in Appendix A, the transmission factor increases when the thickness of the sample decreases. It is well-known that the transparency of amorphous polymers increases the scatter of the laser light, which in turn increases the effective beam diameter [54]. For the CF/PEEK, the transmission was 0% over an entire wavelength range. This is due to the high absorption ability of the carbon fibres. Chabert et al. [16] reported that below 23% of transmission, the power rate of the laser reaching the interface was too low. The energy density at the interface was not high enough to allow polyamide to reach its melting temperature. In Figure 10b, it is observed that the thermal dependence of the transmittance of PEI was negligible in the range of 20 °C to 140 °C. Since there is no thermal transition, it is expected the same transmittance for PEI upon heating up to T_g_ = 217 °C. For a through transmission laser welding, the upper substrate must be semi-transparent to the laser wavelength, whereas the lower substrate must be opaque. Thus, the beam energy needs to be completely absorbed by the lower substrate which will heat up, and by conduction, will cause the temperature of the upper substrate to rise. According to the transmission factors, PEI plates could be assembled as an upper element by a through transmission laser welding whatever their thickness up to 4 mm. CF/PEEK is suitable as a lower element.

### 3.2. Interfacial Strength of Assemblies

#### 3.2.1. Monitoring of the Interfacial Temperature

To ensure a satisfactory interfacial adhesion, it is necessary to reach a temperature above the melting temperature of PEEK or the glass transition of PEI. Controlling temperature at the interface during welding, therefore, would ensure the quality of the welds. Since temperature is a key parameter for optimizing welding processes, the temperature at the interface during heating and cooling was monitored.

The temperature at the interface was measured during welding for the three processes. For ultrasonic welding, K-type thermocouples were inserted inside the PEI energy director film. The results were reported in Bonmatin’s work [27].

For induction welding, N-type thermocouples were placed at the start and end of the weld, the results are shown in Figure 11.

The maximum temperature reached for welding CF/PEEK on CF/PEEK was higher than the melting temperature of PEEK. It took about 30 s to achieve the maximum temperature at 375 °C (Figure 11a). Besides, a lower temperature of 300 °C was sufficient for welding PEI on CF/PEEK (Figure 11b), which was higher than the glass transition of PEI at 216 °C. This result is consistent with our hypothesis explaining that PEI chains facilitate the motion and the diffusion of PEEK across the interface. Thus, less energy is required to soften the interface.

During transmission laser welding, three K-type thermocouples were placed in the PEI (upper element) close to the interface, at the start, in the middle, and at the end of the weld.

Even though temperature monitoring is recognized as essential to improving the quality of welds, in-situ measurements are rare in the literature. The obstacles come from the closed interface and the displacement of the contact surface in the case of ultrasonic welding. The main approach consists of embedding a thermocouple inside an interfacial polymer layer [27] or inside a part to be welded as close as possible to the contact surface [55]. Another option is to measure the temperature distribution by infrared thermography with the camera field either perpendicular [17] or parallel to the welded interface [56,57,58,59]. Then, fiber Bragg grating sensors are constructed in a short segment of optical fibre that reflects particular wavelengths of light and transmits all others. They are used to monitor physical parameters such as temperature even in inaccessible, unconventional environments. One such application is monitoring the temperature of a substrate from the back surface such as any conventional thermocouple sensor [60,61]. In the latter case, numerical models were applied to calculate the interfacial temperature taking into account the thermal properties of materials [62].

For welding processes hereby considered, temperature ramps measured with thermocouples are gathered in Table 1.

Thus, the fastest temperature increase was for ultrasonic welding. Whatever the process, we assume the maximum temperature at the interface reached the thermal transitions of PEI and PEEK as defined by DSC without exceeding the degradation temperature determined by TGA. This assumption was based on an interfacial resistance obtained from mechanical tests presented below.

We underline that such degradation temperature was measured in slow ramps (5 K·min^−1^). When PEEK and CF/PEEK are exposed to rapid heating rates, they withstand much higher temperatures up to 600 °C without evidence of degradation, as attested by Gaitanelis [48] and Bonmatin’s work [27].

#### 3.2.2. Mechanical Resistance of Welded Parts

The resistance of welded specimens was probed by single lap shear tests. The strength (LSS) values are reported in Figure 12. Three configurations were explored as reported in Figure 2, Figure 3 and Figure 4 for an ultrasonic, induction and laser welding respectively. For ultrasonic and laser processes, the assemblies were obtained by overlapping two 25 mm width specimens. For induction welding, two 50 mm width specimens were overlapped to fit our induction welding set-up. Then, they were cut to provide the 25 mm width assemblies. Some of them were separated during cutting, which reflected the low resistance of the welds: these specimens were discarded. The nomenclature is either material X/PEI/material Z when a PEI film was placed at the interface or material X/material Z without an interfacial film. For example, HM63PEEK/PEI/HM63PEEK refers to a joint between two elements of CF/PEEK with a PEI energy director. At least 6 assemblies welded with identical parameters were tested. Each point of the graphs corresponds to an assembly, whereas the bars are the average value of these points, with the standard deviation.

LSS was calculated from Equation (2) by dividing the force to separate the welded specimens by the welded area. For ultrasonic welding, the welded areas were considered to be the fully overlapped surface: 25 × 10 = 250 mm^2^. When observing the fractured surface, the overlapped surface looked totally melted, which strengthened our choice. Oppositely, the welded areas obtained by induction and laser transmission were theoretically 25 × 20 = 500 mm^2^. However, because the melted zone did not correspond to the entire surface, each surface was measured precisely to ensure a reliable LSS value. The welded areas obtained by an induction and laser transmission were calculated using the software ImageJ on PEI surfaces. The areas could not be measured on composite surfaces because the melted zones were not clearly visible on black surfaces. In the case of induction, the size of the welded areas was from 160 mm^2^ to 290 mm^2^ for PEI on CF/PEEK assemblies and from 460 mm^2^ to 520 mm^2^ for CF/PEEK on CF/PEEK assemblies, meaning that 32% to 100% of the overlapped surface was effectively welded. Laser welding provides a very repetitive bonded area. The geometries were two parallel lines apart to create one large welded zone. The width of one line was estimated between 4 mm and 4.5 mm, whereas the distance of laser passage between them was set up at 6 mm. The variability of the welded surfaces was very low, with an average value of 210 ± 5 mm^2^ corresponding to 42% of the overlapped surface, regardless of the thickness of the upper substrates. This highlights the reproducibility of the welds produced by a through transmission laser welding.

In Figure 12a,b, the values obtained for PEI on CF/PEEK assemblies correspond to a 1 mm thick PEI substrate. In addition, a PEI film was used as an energy director in ultrasonic welding for CF/PEEK on CF/PEEK welds. In the case of laser welding, 1, 2, and 4 mm thick PEI substrates were assembled on 1 mm thick CF/PEEK.

For some 1 and 2 mm thick PEI on CF/PEEK assemblies, the fracture took place in the PEI specimen instead of at a welded interface. Therefore, the LSS results were not the resistances of the weld but those of the PEI substrate. These values indicate that the weld was stronger than the maximum stress acceptable by the PEI plates. This case is discussed below. The highest LSS mean value was noticed for ultrasonic welding for welds between two CF/PEEK composites at 22 MPa. The best connection between the PEI and CF/PEEK was obtained for transmission laser welding with 2 mm thick PEI at a mean value of 13 MPa. With the same configuration, the LSS obtained for ultrasonic welding was 7 MPa. For both configurations, the lowest values of LSS were obtained for induction welding with an average of 10 MPa and 4 MPa for CF/PEEK on CF/PEEK and PEI on CF/PEEK, respectively.

The narrowest standard deviation was obtained for laser welding. The highest standard deviation was for induction welding when joining CF/PEEK on CF/PEEK and for an ultrasonic welding when joining CF/PEEK on CF/PEEK with a PEI energy director. However, the highest LSS values were obtained with the latter process/material couple with about 30 MPa. This result could stem from the difficulty to prevent the PEI interfacial layer from flowing. Indeed, a reasonable flow is beneficial to the mechanical resistance, whereas an excessive flow of PEI induces a fibre displacement and deformation of the specimens, resulting in a lower strength.

To sum up, ultrasonic welding seems to be well-suited for assembling composite on composite whereas transmission laser welding is appropriate for PEI/composite welds.

The analysis of fracture modes is presented below based on photographs of assemblies after the mechanical tests. The images of fracture surfaces for an ultrasonic, induction, and laser welding are presented in Figure 13, Figure 15 and Figure 16, respectively.

Ultrasonic welding

For ultrasonic welding, in Figure 13a, a slight fibre distortion at the edges of the CF/PEEK on CF/PEEK welds (with a PEI film at the interface) was observed, the latter could be due to the welding load of 500 N applied up to the cooling. A significant flow of the PEI was observed with slight deformation of the fibres on the edges of the assemblies. A mixed cohesive/adhesive failure occurred at the interface and in the inner layers of the two substrates. In Figure 13b, the welding of PEI on CF/PEEK resulted in a break in the PEI substrate. This indicates that the resistance of the interface was greater than that of the PEI substrate, whose thickness was 1 mm in this case.

For further analysis, Figure 14 displays the interfacial fracture of a weld that exhibits both cohesive and adhesive failure within CF/PEEK plies. In Figure 14a, the optical micrograph from an entire welding area after a lap shear test is presented. An image of a specific location for two CF/PEEK plies is presented in Figure 14b. In that case, the fibres were pulled out, confirming that a cohesive fracture within one ply occured. Some fibres from the second ply were attached to the second composite specimen which is obvious from the height difference of 150 µm measured by optical microscopy. The layer on the right belongs to the first ply of the composite of 140 µm whereas the layer on the left belongs to the second one. A hypothesis is that cracks initiated in both CF/PEEK overlapped specimens of this assembly. The cracks crossed from one CF/PEEK layer to the next one through the PEI-rich interphase, giving rise to this rough surface.

With the PEI film at the interface, the fractured surfaces show PEI flowing along the edge of a transverse outer lap. Also, a fibre breakage in the middle of the lap was observed. Further examination of the damaged areas on the fracture surface revealed the presence of dry carbon fibres on the edge of the transverse outer lap, which could be interpreted as a sign of thermal degradation. It is possible that the maximum temperature reached was locally too high.

Induction welding

In the case of induction welding in Figure 15a, neither a fibre distortion nor polymer flow was noticed, ensuring accurate dimensional stability. For CF/PEEK on CF/PEEK, a mixed cohesive/adhesive failure occurred at the interface and the inner layers of both substrates. Because of black surfaces and slightly deformed samples, determining the welded area was not possible. For this reason, we considered the total overlapped area to calculate LSS, which underestimates the LSS value.

Welds of PEI on CF/PEEK in Figure 15b allow observing a presence of bubbles or a grainy appearance on some specimens. This could come from a material flow during the application of pressure up to 1.5 MPa. Half of the 1 mm thick PEI assemblies broke within PEI, indicating that an interface resistance was higher than that of the PEI substrate. The others gave rise to an adhesive rupture, with the presence of a few fibres adhered to the CF/PEEK substrate.

Through transmission laser welding

In Figure 16, the failure of the PEI substrate is observed for the assemblies with the 1 mm and 2 mm thick PEI. Again, this implies that the interface had an LSS higher than the resistance of the PEI substrate. In the case of the 4 mm thick PEI, a rupture of the interface was observed. Some fibres of the CF/PEEK remained on the PEI, indicating a cohesive failure.

Still, for laser welding, Figure 17 shows an optical micrograph from the entire welding area after a lap shear test. In both cases, it is possible to observe some fibres on the surface of the PEI substrate, proving a strong adhesion at the interface. The differences in failure and LSS results are significant: the average LSS was higher for 2 mm thick PEI than for 4 mm thick. Indeed, a weld of 4 mm PEI on CF/PEEK in Figure 17b looks degraded with a more uneven and darker surface compared to 2 mm PEI. Such thermal degradation may be caused by changes in energy density applied during the welding process. Indeed, for welds of 2 mm PEI on CF/PEEK and 4 mm PEI on CF/PEEK, the energy density of 14.7 J·mm^−2^ and 19.5 J·mm^−2^ were used, respectively. The energy density was increased to compensate for the decrease in transmission factor when the PEI is thicker, as seen in Section 3.1 in Figure 10. However, too high energy density may provoke the degradation of the material and result in lower mechanical properties.

Summary

For PEI on CF/PEEK welds, the failure of PEI was observed for all welds whatever the welding process. Indeed, the PEI specimens were solicited in a tensile mode and when the load reached a tensile yield, the PEI specimen broke. This means that the welded joint was stronger than the PEI. The tensile properties of injection moulded PEI specimens were measured: the ultimate tensile strength was 99 MPa and the elastic modulus was 3300 MPa. The results obtained when the PEI substrate failed were determined by the tensile stress experienced by the PEI substrate. Considering the maximum force reached during the LSS test divided by the specimen section (width × thickness), an average value of 50 MPa was obtained; half of the ultimate tensile strength (UTS) of injection moulded PEI specimens. One could be noticed that the welded specimens were manufactured by compression moulding whereas UTS was obtained from injection moulded specimens. It is well-known that injection moulding orientates the macromolecules in the flow direction; the latter is also the testing direction. When pulled out, the intramolecular bonds—mainly C-C covalent bonds—are stretched. Oppositely, in compression moulding, the polymer conformation obeys the Gaussian random coil: there is no preferential orientation in their organization. During LSS testing, the intermolecular bonds—van der Waals and hydrogen bonds, whose dissociation energy is much lower—are also involved in the mechanical resistance.

Amanat et al. [33] reported different failure modes: interfacial, bulk substrate, near interfacial substrate, substrate, and interfacial. The mode of failure for joints was adapted from classifications used with adhesive-based joints [63,64]. In our case, the failure in the PEI substrate resulted in a tensile yield and break (Figure 13b, Figure 16a,b). When the thickness was 4 mm (Figure 16c), the PEI did not break, highlighting the weld resistance.

During ultrasonic welding, a PEI film inserted between the two CF/PEEK specimens strengthens the welds. Indeed, when assembling CF/PEEK on CF/PEEK, some fibre/fibre contact is possible in some areas, preventing a full close contact. Adding a PEI film ensures a polymer layer that is softened upon heating. However, it seems that the pressure was so high that this layer flowed beyond the specimen edges, as a consequence, dimensional accuracy was not kept.

On the contrary, dimensional accuracy was guaranteed for induction welding. However, the fracture surfaces between composites (Figure 17) showed damage in the CF/PEEK material along the transverse external overlap edge and also in the middle of the overlapped zone. A closer look into the damaged areas on the fracture surface revealed the presence of dried carbon fibres (without polymeric matrix) at the transverse external overlapped edge, which could be interpreted as a sign of thermal degradation. The middle of the overlap shows, however, fractures within a resin-rich interlayer.

## 4. Conclusions

The originality of this work is to compare ultrasonic, induction and transmission laser welding processes for the assembly of thermoplastic composites. The targeted application is a skin for sandwich panels for the space industry. The process parameters have still to be optimized, however, selected operation points give values of LSS high enough to fit some industrial specifications.

The materials were thoroughly characterized to highlight the specific properties involved in each welding process. Laser welding requires the upper element to be transparent to the laser wavelength. The transmission factor of PEI was 52%, 50%, and 43% at 20 °C for 1, 2, and 4 mm thick samples at the wavelength of interest at 970 nm. Induction welding required the material to be electrically and thermally conductive. The thermal conductivity of CF/PEEK was 55 W·(m·K)^−1^, which makes it suitable to be welded by induction without a metallic susceptor. Ultrasonic welding is the most versatile process as it does not necessitate any material properties.

For CF/PEEK on CF/PEEK, the maximum LSS value of 28.6 MPa was reached for an ultrasonic welding, while an induction one brought 17.6 MPa. This difference is partially due to the addition of a 250 µm thick PEI film used as an interfacial energy director for ultrasonic welding. CF/PEEK on CF/PEEK joints resulted in mixed cohesive/adhesive failure at the interface and within the inner layers of both substrates.

For PEI on CF/PEEK assemblies, the three processes are suitable. The maximum LSS values were obtained for laser welding at 15.2 MPa, followed by ultrasonic welding at 8.5 MPa and induction welding at 4.2 MPa. These low values are explained by the fracture of PEI specimens when the PEI thickness was below 4 mm. The LSS values reported are not associated with the interfacial resistance but with the tensile strength of the PEI itself. The highest LSS for welds with PEI were obtained by laser welding for PEI of 2 mm thickness. It is worth highlighting the effect of the low thermal conductivity of PEI compared to CF/PEEK layers. The heat is dissipated faster inside the composite layers due to carbon fibres, which require reaching higher temperatures to bring enough heat to the interface in the case of CF/PEEK on CF/PEEK assemblies.

In all welding processes, controlling the interfacial temperature is not easily controlled. The temperature must stay above the glass transition of PEI or melting temperature of PEEK while preventing degradation and delamination of composite plies. Further research efforts are necessary to develop reliable in-situ temperature measurements to monitor an interfacial temperature, as a step towards dimensional accuracy and weld resistance. Further works will be oriented to gain insights into the effect of process parameters on temperature distribution in the contact area.

Finally, the three processes are promising for assembling carbon fibre/PEEK composites. The ongoing study aims to optimize the process parameters to increase the strength of the welds, before transferring the processes for assembling parts for space applications such as satellites.

## Figures and Tables

**Figure 1 materials-15-06365-f001:**
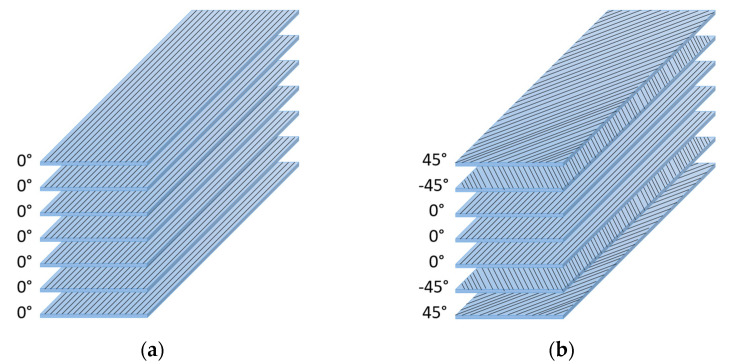
Sequence of the CF/PEEK films (**a**) unidirectional and (**b**) cross stacking.

**Figure 2 materials-15-06365-f002:**
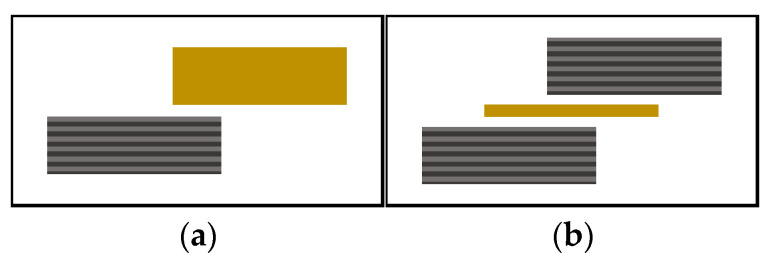
Single-lap welded joints for USW (**a**) welding of PEI on CF/PEEK and (**b**) welding of CF/PEEK on CF/PEEK with 250 µm thick PEI as energy director (yellow bars are for PEI, grey bars for CF/PEEK).

**Figure 3 materials-15-06365-f003:**
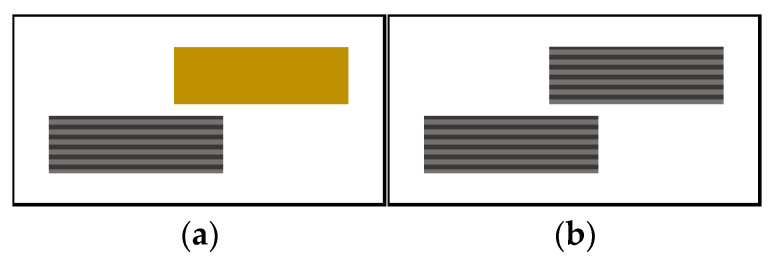
Single-lap welded joints for IW (**a**) welding of PEI on CF/PEEK and (**b**) welding of CF/PEEK on CF/PEEK (yellow bar is PEI, grey bars for CF/PEEK).

**Figure 4 materials-15-06365-f004:**
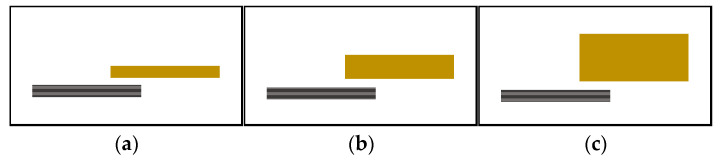
Single-lap welded joints for LW (**a**) welding of PEI 1 mm thick on CF/PEEK, (**b**) welding of PEI 2 mm thick on CF/PEEK and (**c**) welding of PEI 4 mm thick on CF/PEEK (yellow bars are PEI, grey bars for CF/PEEK).

**Figure 5 materials-15-06365-f005:**
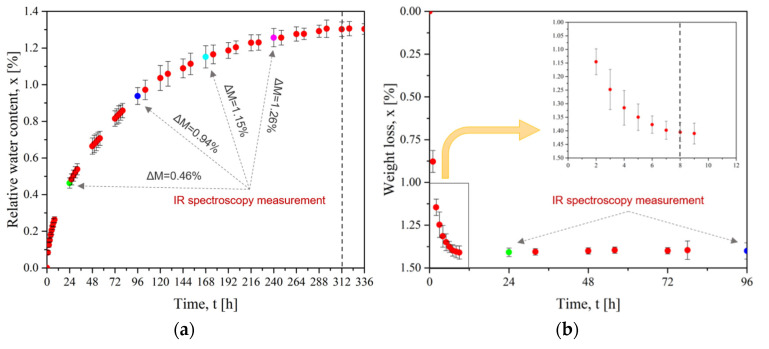
For PEI (**a**) Evolution of water diffusion at 25 °C and (**b**) Evolution of mass loss with drying time at 150 °C (dots of different colors represent measurement by FTIR spectroscopy).

**Figure 6 materials-15-06365-f006:**
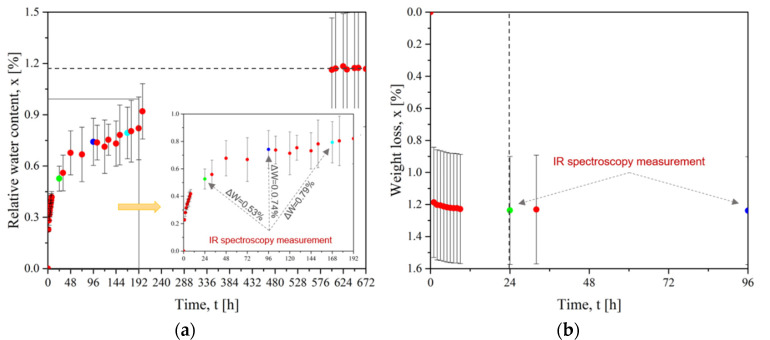
For HM63/PEEK (**a**) Evolution of water diffusion with time at 25 °C and (**b**) Evolution of mass loss with drying time at 150 °C (dots of different colors represent measurement by FTIR spectroscopy).

**Figure 7 materials-15-06365-f007:**
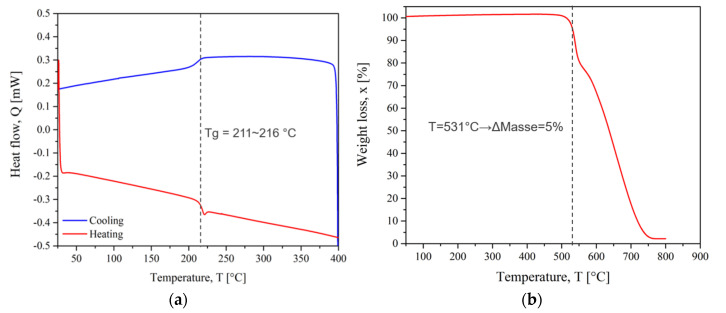
PEI thermograms (**a**) DSC and (**b**) TGA.

**Figure 8 materials-15-06365-f008:**
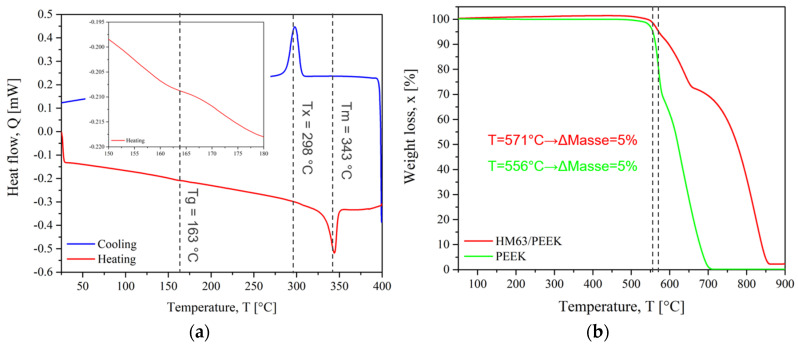
CF/PEEK thermograms (**a**) DSC and (**b**) TGA.

**Figure 9 materials-15-06365-f009:**
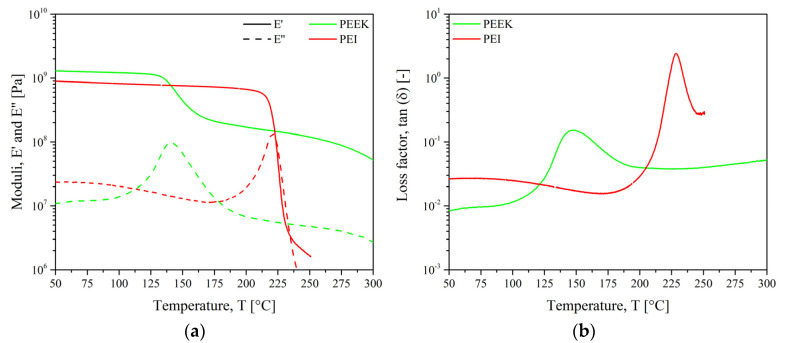
Dynamic mechanical analysis for pure PEEK and PEI (**a**) storage and loss moduli and (**b**) loss factor.

**Figure 10 materials-15-06365-f010:**
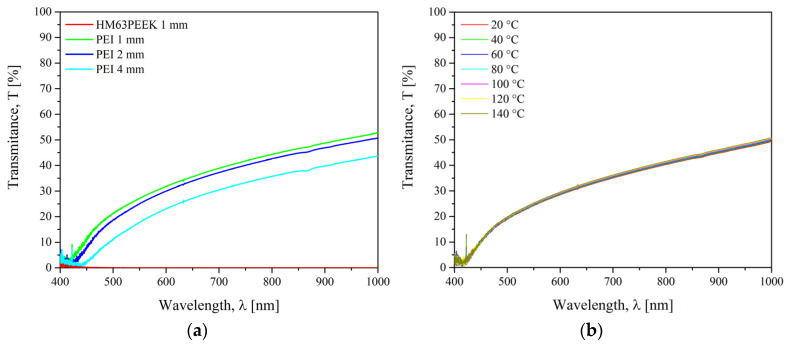
Transmission coefficient of (**a**) PEI and HM63/PEEK at 20 °C and (**b**) PEI for different temperatures.

**Figure 11 materials-15-06365-f011:**
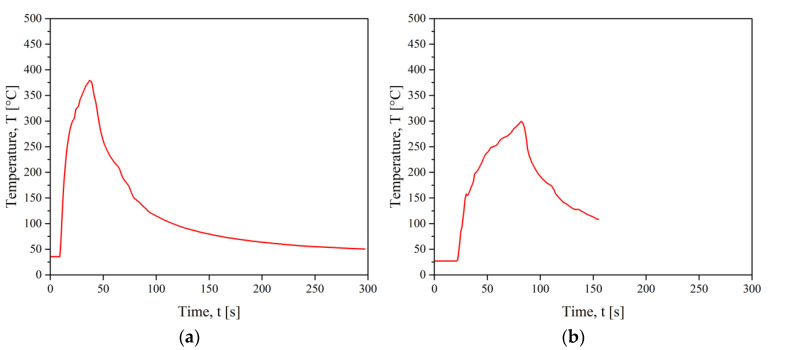
Temperature monitoring during IW for (**a**) CF/PEEK on CF/PEEK and (**b**) PEI 1 mm on CF/PEEK.

**Figure 12 materials-15-06365-f012:**
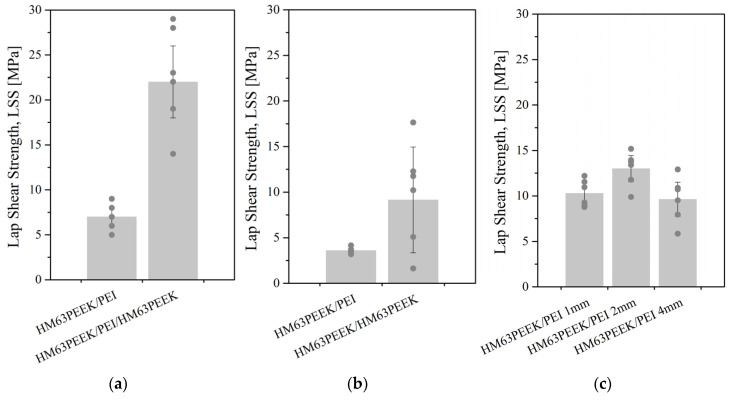
Lap shear strength of test specimens after (**a**) ultrasonic, (**b**) induction and (**c**) transmission laser welding. LSS is calculated from the effective welded surface.

**Figure 13 materials-15-06365-f013:**
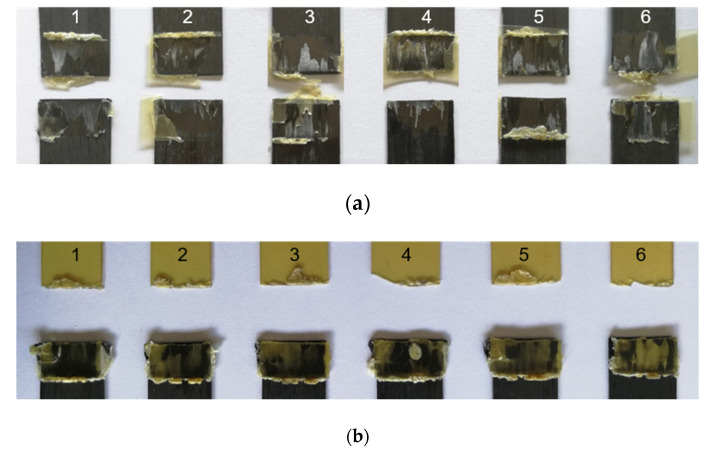
Fracture surfaces after LSS testing of ultrasonically welded specimens from 1 to 6 for (**a**) CF/PEEK on CF/PEEK with 250 µm thick PEI as energy director and (**b**) 2 mm thick PEI on CF/PEEK.

**Figure 14 materials-15-06365-f014:**
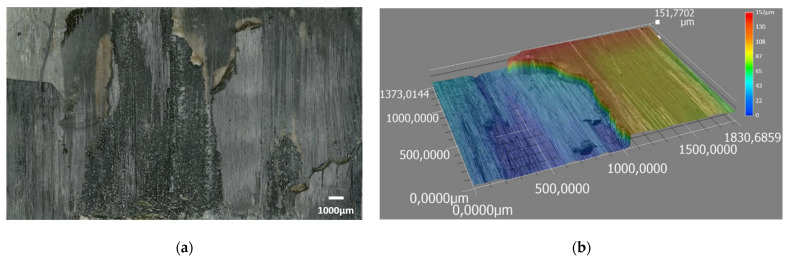
Optical micrographs after LSS testing of ultrasonically welded of (**a**) interfacial fracture of two layers of CF/PEEK and (**b**) colour chart corresponds to a height difference.

**Figure 15 materials-15-06365-f015:**
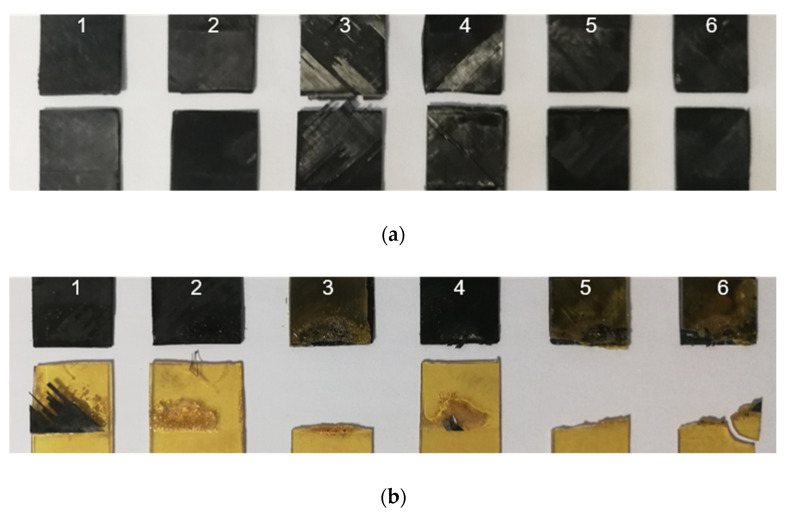
Fracture surface after LSS testing of induction welded specimens from 1 to 6 for (**a**) CF/PEEK on CF/PEEK and (**b**) 1 mm thick PEI on CF/PEEK.

**Figure 16 materials-15-06365-f016:**
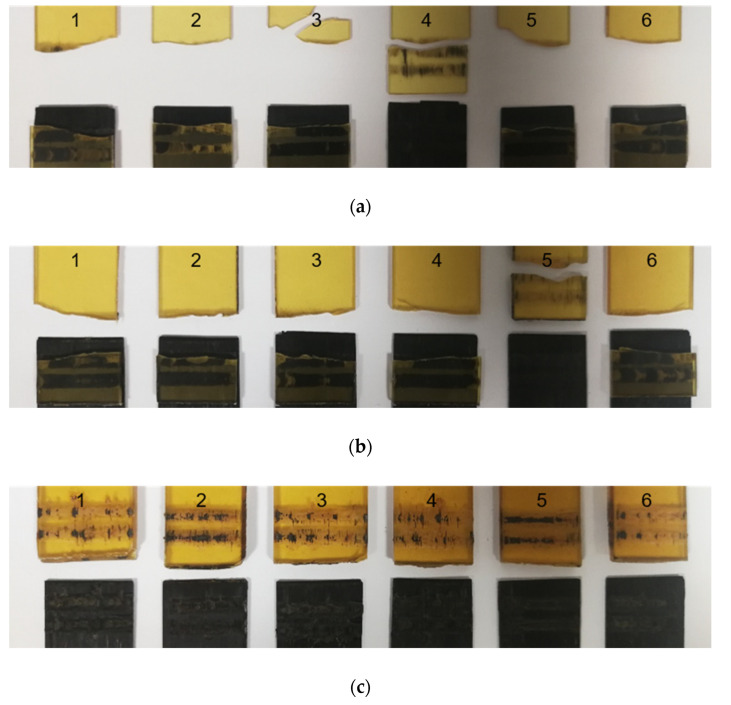
Fracture surface after LSS testing of transmission laser welded specimens from 1 to 6 for (**a**) 1 mm PEI, (**b**) 2 mm PEI and (**c**) 4 mm thick PEI on CF/PEEK.

**Figure 17 materials-15-06365-f017:**
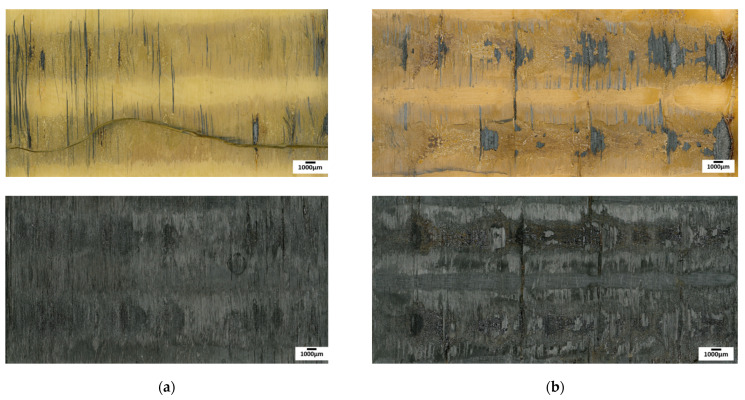
Optical micrographs of interfacial fracture after LSS testing of transmission laser welded of (**a**) 2 mm PEI and (**b**) 4 mm thick PEI on CF/PEEK (on the top, transparent PEI, on the bottom, absorbent CF/PEEK).

**Table 1 materials-15-06365-t001:** Heating and cooling temperature ramps for welding processes.

In K·s^−1^	Ultrasonic Welding	Induction Welding	Laser Welding
Heating ramp	1000	10	200
Cooling ramp	500	5	100

## Data Availability

The data that support the findings of this study are available from the corresponding author.

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
