# Peer review of "Assembling of Carbon Fibre/PEEK Composites: Comparison of Ultrasonic, Induction, and Transmission Laser Welding"

_materials, 2022, doi:10.3390/ma15186365_

Round 1
Reviewer 1 Report
1. For the title, carbon/PEEK composites should be changed to carbon fiber/PEEK composites. The authors should note that the term carbon should be broader and must include graphene, carbon nanotubes, etc., which is beyond the focus of this manuscript.
2. For the abstract, the two paragraphs should be combined into one. In addition, the significance and innovation of the main findings of this work should be pointed out at the end of the paragraph.
3. For the introduction, the corresponding text description should be shortened. Because this paper is a research paper (not a review), the authors should review the research progress directly related to this paper and give detailed literature, rather than providing an overly long background introduction.
4. The unit writing of some physical quantities in the manuscript is not standard, and the author should pay attention to the difference between the period (.) and the multiplication symbol (·). For example the following:
(1) In Line189-202,°C.min-1 should be °C·min-1.
(2) In Line358, N.min-1 should be N·min-1.
(3) In Line375, mm.s-1 should be mm·s-1.
(4) ……
5. Writing something like Figure 5.a is not recommended, it should be written as Figure 5a. Authors should note that similar writing in the full text (not noted by the reviewer) should be revised.
6. In the field of thermal analysis, the crystallization temperature is usually represented by Tx, and Tc is mostly used for the Curie temperature. It is suggested that the authors change the crystallization temperature Tc to Tx in the text.
7. In actual production, it is impossible to install thermocouples at the interface. What monitoring method does the author have or what process conditions (welding time, pressure?) should be met to avoid excessive temperature rise?
8. It appears that the performance degradation of PEI in welding plays a significant role in welding quality. How to ensure that the mechanical properties of PEI are not degraded during the welding process? How about brazing?
9. The conclusion part is too long. It is recommended that the author delete some of the content and give the main results, innovation and research significance.
Author Response
Dear Editor,
we are sincerely grateful to the reviewers for considering our work and for their useful comments on the original research article entitled “Assembling of carbon fibre/PEEK composites: comparison of ultra-sonic, induction and transmission laser welding” by Adrian Korycki, Christian Garnier, Margot Bonmatin, Elisabeth Laurent and France Chabert, for consideration for publication in Materials (Manuscript ID: materials-1884876).
Our responses are presented below in blue. Hopefully, these answers will clarify the reviewers’ points. Some changes were done in the attached manuscript; they are highlighted in yellow.
The authors would like to thank the reviewers for their valuable comments that improved the quality of the manuscript. Below we address them one by one. The authors have done their best to improve the quality of the manuscript to meet the high standard of the journal
We would like to thank the reviewers for the thorough review. We have gone through the comments one by one and addressed them as detailed below. We also improved the paper to meet the standard expected by the reviewers. Special attention was given to the abstract and introduction section by rewording some sections as suggested by the reviewer and clarifying some points. We have also clarified the novelty issues claimed.
Reviewer #1 (16/08/2022):
For the title, carbon/PEEK composites should be changed to carbon fiber/PEEK composites. The authors should note that the term carbon should be broader and must include graphene, carbon nanotubes, etc., which is beyond the focus of this manuscript.
Response: Thank you for your remark. We agree with you; the title has been corrected with carbon fibre/PEEK instead of carbon/PEEK.
For the abstract, the two paragraphs should be combined into one. In addition, the significance and innovation of the main findings of this work should be pointed out at the end of the paragraph.
Response: Thank you, we have modified the abstract to fit the reviewer’s expectations.
For the introduction, the corresponding text description should be shortened. Because this paper is a research paper (not a review), the authors should review the research progress directly related to this paper and give detailed literature, rather than providing an overly long background introduction.
Response: Thank you for your comment. The introduction section has been fully reviewed and shortened. We also added some new references corresponding to induction and transmission laser welding processes and removed other ones whose relevance was weaker.
The unit writing of some physical quantities in the manuscript is not standard, and the author should pay attention to the difference between the period (.) and the multiplication symbol (·). For example, the following:
In Line189-202,°C.min-1 should be °C·min-1.
In Line358, N.min-1 should be N·min-1.
In Line375, mm.s-1 should be mm·s-1.
Response: Thank you for the comment. We have replaced all the dots (.) by the multiplication symbol (·) in all the units.
Writing something like Figure 5.a is not recommended, it should be written as Figure 5a. Authors should note that similar writing in the full text (not noted by the reviewer) should be revised.
Response: Thank you for the comment. All the Figures names have been corrected.
In the field of thermal analysis, the crystallization temperature is usually represented by Tx, and Tc is mostly used for the Curie temperature. It is suggested that the authors change the crystallization temperature Tc to Tx in the text.
Response: Thank you for your suggestion. The crystallization temperature has been changed to Tx in the text and on the Figure 8a.
In actual production, it is impossible to install thermocouples at the interface. What monitoring method does the author have or what process conditions (welding time, pressure?) should be met to avoid excessive temperature rise?
Response: Thank you for your question. As the interface is a closed contact during welding, measuring the temperature at the interface is tricky. In the present work, thermocouples have been inserted as close as possible to the interface to measure the temperature during welding for the three processes. The experimental method is described in section 3.2.1 Monitoring of the interfacial temperature. For ultrasonic welding, more results are displayed in ref. (27) from the same authors. We are aware of the difficulty to transfer this method of measurement to production. Instead, we aim to improve the knowledge to understand the effect of the parameters on the temperature ramps. Once the effect of parameters will be fully mastered, it will be simpler to monitor the process through parameters such as time, specimen displacement, power or pressure. Like other authors, we contribute to improve this knowledge.
It appears that the performance degradation of PEI in welding plays a significant role in welding quality. How to ensure that the mechanical properties of PEI are not degraded during the welding process? How about brazing?
Response: We agree with the reviewer, adding a PEI layer to assemble CF/PEEK specimens is comparable to brazing of metals which consists of adding a filler metal whose melting point is lower. So, the filling material must withstand high temperatures involved during welding. We have checked the thermal stability of PEI by TGA (see Figure 7). Moreover, we have registered the FTIR spectra of PEI before and after welding (not shown). No significant change was noticed, so we assume that PEI is not degraded during welding. In the literature, two studies are relevant to confirm our hypothesis: the first one (Augh and Gillespie, Degradation of Continuous Carbon Fiber Reinforced Polyetherimide Composites during Induction Heating, Journal of THERMOPLASTIC COMPOSITE MATERIALS, 14 (2001)) relates no changes in molecular weight, glass transition temperature, or mechanical properties of PEI during induction process. Because the time scale in the induction heating process is so short, almost no degradation was observed. The other one (Amancio Filho et al., Thermal degradation of polyetherimide joined by friction riveting (FricRiveting). Part I: Influence of rotation speed, Polymer Degradation and Stability, 93, 1529–1538 (2008)) indicates that the primary thermal degradation mechanism of PEI is chain scission, with a very small decrease in average molecular weight. This minor decrease in molecular weight can be considered irrelevant for the mechanical performance of the joints because it is either above or within the molecular weight range where the strength of the PEI is independent of the variations in this property.
This short part has been added in the section to reinforce our thoughts:
“Augh and Gillespie (Augh and Gillespie, Degradation of Continuous Carbon Fiber Reinforced Polyetherimide Composites during Induction Heating, Journal of THER-MOPLASTIC COMPOSITE MATERIALS, 14 (2001)) relates no changes in molecular weight, glass transition temperature, or mechanical properties of PEI during induction process. Because the time scale in the induction heating process is so short, almost no degradation was observed. Amancio Filho et al. (Amancio Filho et al., Thermal degradation of polyetherimide joined by friction riveting (FricRiveting). Part I: Influence of rotation speed, Polymer Degradation and Stability, 93, 1529–1538 (2008)) indicates that the primary thermal degradation mechanism of PEI is chain scission, with a very small de-crease in average molecular weight. This minor decrease in molecular weight can be considered irrelevant for the mechanical performance of the joints because it is either above or within the molecular weight range where the strength of the PEI is independent of the variations in this property. So, we assume that the PEI macromolecular structure is not significantly modified by the welding cycles.”
The conclusion part is too long. It is recommended that the author delete some of the content and give the main results, innovation and research significance.
Response: Thank you, the conclusion has been shortened and some sentences have been summarized to clarify the main findings.
Reviewer 2 Report
1. “because the melted zone does not correspond to the entire surface, each surface is measured precisely to ensure a reliable lap shear strength (LSS) value. ”------Does it mean that each specimen has different overlapped surface? If so, how to guarantee the required overlapped surface in practical applications?
2. “determining the welded area was not possible. For this reason, we considered the total overlapped”----How to ensure the designed (required) LSS in practice?
3. “the degradation of the material by heat propagation”---how much reduction of the ultimate tensile strength of the substrate is after welding? The information should be provided for the cases when the interface has an LSS higher than the resistance of the substrate.
4. A few abbreviations in abstract and main text have not been defined. The definition should be given when they appear for the first time.
5. “yellow bars”? In Fig. 3, only one yellow bar is there. Delete “s” in bars.
6. “due to the pressure of 500 N”; “of pressure up to 15 bars”----please check the unit of pressure. Besides, SI unit should be used.
7. “When propagating from one specimen to another, the cracks go ”---how can the crack propagate from one specimen to another?
8. English should be polishing. Tense should be consistent. Authors should read the entire manuscript carefully to eliminate the grammar errors and misprints. For instance, “Glass transition temperature (Tg), crystallization temperature (Tc) and melting temperature (Tm) was obtained ”; “The highest LSS for welds with PEI were obtained”; etc.
Author Response
Dear Editor,
we are sincerely grateful to the reviewers for considering our work and for their useful comments on the original research article entitled “Assembling of carbon fibre/PEEK composites: comparison of ultra-sonic, induction and transmission laser welding” by Adrian Korycki, Christian Garnier, Margot Bonmatin, Elisabeth Laurent and France Chabert, for consideration for publication in Materials (Manuscript ID: materials-1884876).
Our responses are presented below in blue. Hopefully, these answers will clarify the reviewers’ points. Some changes were done in the attached manuscript; they are highlighted in yellow.
The authors would like to thank the reviewers for their valuable comments that improved the quality of the manuscript. Below we address them one by one. The authors have done their best to improve the quality of the manuscript to meet the high standard of the journal
We would like to thank the reviewers for the thorough review. We have gone through the comments one by one and addressed them as detailed below. We also improved the paper to meet the standard expected by the reviewers. Special attention was given to the abstract and introduction section by rewording some sections as suggested by the reviewer and clarifying some points. We have also clarified the novelty issues claimed.
Reviewer #2 (16/08/2022):
“because the melted zone does not correspond to the entire surface, each surface is measured precisely to ensure a reliable lap shear strength (LSS) value.” Does it mean that each specimen has different overlapped surface? If so, how to guarantee the required overlapped surface in practical applications?
Response: Each assembly is made with overlapping two specimens. Whatever the process, a pressure is applied to insure intimate contact and polymer diffusion of the both. This pressure can modify the overlapped surface. Instead of considering the theoretical contact area, we choose to measure this surface to improve the reliability of the LSS values. However, in the case of ultrasonic welding and transmission laser welding, the area difference between specimens are at most 2 %. In practical applications, the specimen dimensions and the process parameters (including pressure) are constant, so the overlapped surface is expected to be constant.
“determining the welded area was not possible. For this reason, we considered the total overlapped” How to ensure the designed (required) LSS in practice?
Response: Thank you for the question. The LSS value of an assembly depends on many factors: initial surface roughness, rheological behaviour, entanglement rate, degree of crystallinity, which depend on process parameters such as temperature, pression and time. When the pressure/time/temperature are high, the intimate contact rate will reach the maximum and the real contact surface could be higher than the theoretical surface because of squeezing of specimens and polymer flowing on the edges. Even if the LSS is high, the assembly will potentially not fulfill the requirement in dimensional accuracy. In practice, a combination of parameters pressure/time/temperature gives a satisfying LSS with dimensional accuracy. This combination of parameters has to be defined by large sets of experiments for each material/welding machine couple.
“the degradation of the material by heat propagation” How much reduction of the ultimate tensile strength of the substrate is after welding? The information should be provided for the cases when the interface has an LSS higher than the resistance of the substrate.
Response: Thank you for the question which has made us take a deeper analysis. It is unlikely that a so big difference comes from the degradation of PEI during welding: the FTIR spectra revealed no change in the chemical structure of the PEI. Instead, we assume that the preparation of the specimens impacts the macromolecular conformation and thus, the mechanical resistance of PEI.
To revise our explanation, the following part has been added in the manuscript:
“Considering the maximum force reached during the SLS test divided by the specimen section (width x thickness), an average value of 50 MPa is obtained; half of the UTS of injection moulded PEI specimens. One could be noticed that the welded specimens were manufactrured by compression molding whereas UTS is obtained from injection moulded specimens. It is well know that injection moulding orientate the macromolecules in the flow direction; the latter is also the testing direction. When pull out, the intramolecular bonds – mainly C-C covalent bonds - are stretched. Oppositely, in compression molding, the polymer conformation obeys the Gaussian random coil: there is no preferential orientation in their organization. During SLS testing, the intermolecular bonds – van der Waals and hydrogen bonds, whose dissociation energy is much lower - are also involved in the mechanical resistance.”
A few abbreviations in abstract and main text have not been defined. The definition should be given when they appear for the first time.
Response: Thank you for your remark. We have added the definitions to unexplained abbreviations.
“yellow bars”? In Fig. 3, only one yellow bar is there. Delete “s” in bars.
Response: Thank you comment. The text has been corrected.
“due to the pressure of 500 N”; “of pressure up to 15 bars” Please check the unit of pressure. Besides, SI unit should be used.
Response: Thank you for the remark. We have corrected the text: “due to the welding load of 500 N” and changed the unit: “of pressure up to 1.5 MPa”
“When propagating from one specimen to another, the cracks go” How can the crack propagate from one specimen to another?
Response: Thank you for the comment. Each assembly is made by overlapping two specimens. For ultrasonic welding, these two specimens are connected to each other by a PEI film as energy director. In Figure 14, the height difference of 150 µm is measured by optical microscopy with carbon fibres from both sides remaining on the fracture surface. We mean that the crack initiates in one CF/PEEK layer, then it crosses the PEI film to propagate within the other specimens. To clarify this point, we have added the following part:
“A hypothesis is that cracks initiate in both CF/PEEK overlapped specimens of this assembly. The cracks go from one CF/PEEK layer to the next one through the PEI-rich interphase, giving rise to this rough surface.”
English should be polishing. Tense should be consistent. Authors should read the entire manuscript carefully to eliminate the grammar errors and misprints. For instance, “Glass transition temperature (Tg), crystallization temperature (Tc) and melting temperature (Tm) was obtained”; “The highest LSS for welds with PEI were obtained”; etc.
Response: Thank you for your suggestion. The paper has been checked by an English teacher in order to remove the grammar mistakes.
Reviewer 3 Report
The paper entitled "Assembling of carbon/PEEK composites: comparison of ultrasonic, induction and transmission laser welding" described the properties of carbon fibers/PEEK composites. It may attract the interest of readers to some extent. But it has many flaws, which should be revised firstly.
1. The title of this manuscript should be revised. The whole discussed carbon fibers. why do you use carbon/PEEK composites in the title?
2. The abstract part should be rewritten. It looks like introduction rather than abstract. The authors should use simple sentences to summarize the meaning of this research. In addition, the important conclusions and experimental data should be well summarized in this part.
3. The authors should well summarize the research background and other researchers' work. Please cite newly references in the introduction part.
4. The logic relation of the paragraphs in the introduction part made me confused. It should be well revised.
5. Can you explain why you choose the sequence of CF/PEEK films rather than other sequence? Why do not you try more sequences so that we can get more comparasion dat?
6. TGA tests should be measured at higher temperature. Because the curve in Fig. 8b was not flat.
7. In Fig. 9, why the curves of PEI sample stopped at 250 oC?
Author Response
Dear Editor,
we are sincerely grateful to the reviewers for considering our work and for their useful comments on the original research article entitled “Assembling of carbon fibre/PEEK composites: comparison of ultra-sonic, induction and transmission laser welding” by Adrian Korycki, Christian Garnier, Margot Bonmatin, Elisabeth Laurent and France Chabert, for consideration for publication in Materials (Manuscript ID: materials-1884876).
Our responses are presented below in blue. Hopefully, these answers will clarify the reviewers’ points. Some changes were done in the attached manuscript; they are highlighted in yellow.
The authors would like to thank the reviewers for their valuable comments that improved the quality of the manuscript. Below we address them one by one. The authors have done their best to improve the quality of the manuscript to meet the high standard of the journal
We would like to thank the reviewers for the thorough review. We have gone through the comments one by one and addressed them as detailed below. We also improved the paper to meet the standard expected by the reviewers. Special attention was given to the abstract and introduction section by rewording some sections as suggested by the reviewer and clarifying some points. We have also clarified the novelty issues claimed.
Reviewer #3 (17/08/2022):
The paper entitled "Assembling of carbon/PEEK composites: comparison of ultrasonic, induction and transmission laser welding" described the properties of carbon fibers/PEEK composites. It may attract the interest of readers to some extent. But it has many flaws, which should be revised firstly.
Response: We would like to thank you for your kind review.
The title of this manuscript should be revised. The whole discussed carbon fibers. why do you use carbon/PEEK composites in the title?
Response: Thank you for your remark. We agree with the reviewer; the title has been corrected with carbon fibre instead of carbon/PEEK.
The abstract part should be rewritten. It looks like introduction rather than abstract. The authors should use simple sentences to summarize the meaning of this research. In addition, the important conclusions and experimental data should be well summarized in this part.
Response: Thank you, the abstract has been revised according to the reviewer’s suggestions.
The authors should well summarize the research background and other researchers' work. Please cite newly references in the introduction part.
Response: Thank you for your comment. The introduction section has been reviewed and shortened. We also added some new references corresponding to induction and transmission laser welding processes.
The logic relation of the paragraphs in the introduction part made me confused. It should be well revised.
Response: Thank you for your remark. The introduction section has been reviewed and some changes in the organization of paragraphs have been applied to better adjust the problematic.
Can you explain why you choose the sequence of CF/PEEK films rather than other sequence? Why do not you try more sequences so that we can get more comparison data?
Response: Thank you for your question. We have tried two different sequences of CF/PEEK films to make our specimens. Based on our previous experience on welding, the unidirectional sequence allows to dissipate heat faster and more efficiently. This configuration is the worse for fusion bonding because most of the generated heat is dissipated on each side of the specimen and heat does not reach the interface to be welded. The cross-stacking sequence is close to industrial cases and it has been adapted to fit induction welding process. For clarifying, we have added the following text:
“the unidirectional sequence is expected to dissipate heat faster and more efficiently. This configuration is the worse for fusion bonding because most of the generated heat is dissipated on each side of the specimen. Higher temperatures must be attained to melt the interface to be welded. The cross-stacking sequence is close to industrial cases and it has been adapted to fit induction welding process.”
TGA tests should be measured at higher temperature. Because the curve in Fig. 8b was not flat.
Response: Thank you for the comment. We did not go further because of the machine limits. However, the interesting temperature was the beginning of degradation with is considered when the polymer loses 5 % of its mass. We do not aim to analyse the degradation mechanisms in this paper.
In Fig. 9, why the curves of PEI sample stopped at 250 °C?
Response: Thank you for your remark. The amorphous plateau is not visible: above 250 °C, the PEI is so soft that measuring is not possible in this configuration: it flows. Instead, a plate/plate configuration would be suitable. Nevertheless, in the range 50-250°C, the glassy plateau and the glass transition are obvious, which is enough for comparing with PEEK.
Round 2
Reviewer 3 Report
The manuscript improved to some extent after revision. But the authors did not address my comments. I think it still need major revision. The detail comments are as follows.
1. Add the TGA curve of pure PEEK, and retake the TGA test of CF/PEEK.
2. There are too much grammar mistakes in the whole manuscript. The experimetnal data which you obtain from the experiment should be described with the past tense.
3. What does LSS mean in the abstract part? Please define the abbreviations when you used them first time. The abstract part still need to be further improved. The language and the conclusions are boring.
4. I did not satify the explanation about my comments "Can you explain why you choose the sequence of CF/PEEK films rather than other sequence? Why do not you try more sequences so that we can get more comparison data?"
5. "TGA tests should be measured at higher temperature. Because the curve in Fig. 8b was not flat." If your intrument can not measure higher temperature. Please go outside and find another TGA to measure the samples. TGA is a very common tool, many universities and institutes have it.
Author Response
The authors would like to thank the reviewer for her/his valuable comments that improved the quality of the manuscript. Below we address them one by one, our responses are presented below in blue. Hopefully, these answers will clarify the reviewer’s points. Some changes were done in the attached manuscript; they are highlighted in green.
Reviewer #3 (31/08/2022):
Add the TGA curve of pure PEEK, and retake the TGA test of CF/PEEK.
Response: Thank you for your remark, we have included the TGA curves within the manuscript as required by the reviewer. This reviewer’s comment is similar than her/his last one, for which we provide a response below.
There are too much grammar mistakes in the whole manuscript. The experimental data which you obtain from the experiment should be described with the past tense.
Response: Thank you for your suggestion. The experimental part has been verified and we have changed the text to the past tense. Depending on journal and author choice, either present or past tense is allowed. However, we have checked all the sentences to change from present to past tense, following the reviewer’s advice.
What does LSS mean in the abstract part? Please define the abbreviations when you used them first time. The abstract part still need to be further improved. The language and the conclusions are boring.
Response: Thank you your remark. LSS means “lap shear strength”, the definition is now included in the abstract. The whole abstract has been reviewed to improve the quality of the writing style and to make it more attractive for the readers.
I did not satify the explanation about my comments "Can you explain why you choose the sequence of CF/PEEK films rather than other sequence? Why do not you try more sequences so that we can get more comparison data?"
Response: Thank you for your question. As explained previously, the unidirectional sequence of plies is chosen as a model system, knowing that the heat dissipation would be higher and faster than for other sequences. The other tested sequence, cross stacking of plies at [45,-45,03,-45,45] brings the suitable mechanical properties to manufacture most of the parts used in aerospace industry. For instance, skins of sandwich panels are produced with this sequence. Could the review give more details on what result she/he expects when changing the sequence? In our opinion, testing another sequence would not bring significant difference on the weld performance : The effect of ply orientation is described in many works such as Rahmani, H., Najafi, S.H.M., Saffarzadeh-Matin, S. and Ashori, A. (2014), Mechanical properties of carbon fiber/epoxy composites: Effects of number of plies, fiber contents, and angle-ply layers. Polym Eng Sci, 54: 2676-2682. https://doi.org/10.1002/pen.23820. Most of these works deal with carbon fiber/epoxy. Similar studies on carbon fiber/PEEK are scarce, because the diffusion of this material is quite new in the industry, its cost is very high and the manufacturing procedure require a expertise to avoid degradation during consolidation. For our purpose, any balanced sequence such as cross-stacking will moderate heat diffusion towards specimen edges to keep the heat concentrated at the interface. Based on our previous experience, we know that only the orientation of the upper ply is of utmost importance, mostly for induction welding, because the heat is generated from the carbon fibers available on the specimen surface. Finally, exploring other sequences would be the topic of a further study as the specimen preparation is very time-consuming and expensive.
"TGA tests should be measured at higher temperature. Because the curve in Fig. 8b was not flat." If your intrument can not measure higher temperature. Please go outside and find another TGA to measure the samples. TGA is a very common tool, many universities and institutes have it.
Response: Thank you your remark. The TGA was carried out up to 900°C for pure PEEK and CF/PEEK in order to obtain a flat line when the degradation is completed. However, we do consider that such tests were not necessary for our study.